# Quantify Piston and Preferential Water Flow in Deep Soil Using Cl⁻ and Soil Water Profiles in Deforested Apple Orchards on the Loess Plateau, China

**Zhiqiang Zhang** [1], **Bingcheng Si** [1,2,3,*] ⓘ, **Huijie Li** [3,*] ⓘ **and Min Li** [1]

[1] Key Laboratory of Agricultural Soil and Water Engineering in Arid and Semiarid Areas, Ministry of Education, Northwest A&F University, Yangling 712100, China; scrzzq@outlook.com (Z.Z.); limin2016@nwafu.edu.cn (M.L.)

[2] Department of Soil Science, University of Saskatchewan, Saskatoon SK S7N-5A8, Canada

[3] School of Resources and Environmental Engineering, Ludong University, Yantai 264001, China

* Correspondence: Bing.Si@usask.ca (B.S.); 3660@ldu.edu.cn (H.L.)

**Abstract:** Piston and preferential water flow are viewed as the two dominant water transport mechanisms regulating terrestrial water and solute cycles. However, it is difficult to accurately separate the two water flow patterns because preferential flow is not easy to capture directly in field environments. In this study, we take advantage of the afforestation induced desiccated deep soil, and directly quantify piston and preferential water flow using chloride ions (Cl⁻) and soil water profiles, in four deforested apple orchards on the Loess Plateau. The deforestation time ranged from 3 to 15 years. In each of the four selected orchards, there was a standing orchard that was planted at the same time as the deforested one, and therefore the standing orchard was used to benchmark the initial Cl⁻ and soil water profiles of the deforested orchard. In the deforested orchards, piston flow was detected using the migration of the Cl⁻ front, and preferential flow was measured via soil water increase below the Cl⁻ front. Results showed that in the desiccated zone, Cl⁻ migrated to deeper soil after deforestation, indicating that the desiccated soil layer formed by the water absorption of deep-rooted apple trees did not completely inhibit the movement of water. Moreover, there was an evident increase in soil water below the downward Cl⁻ front, directly demonstrating the existence of preferential flow in deep soil under field conditions. Although pore water velocity was small in the deep loess, preferential water flow still accounted for 34–65% of total infiltrated water. This study presented the mechanisms that regulate movement of soil water following deforestation through field observations and advanced our understanding of the soil hydrologic process in deep soil.

**Keywords:** preferential flow; piston flow; apple orchard; Loess Plateau; deep soil water

## 1. Introduction

The water in unsaturated zones is the main source of evapotranspiration and groundwater recharge [1–3]. Therefore, clarifying how water migrates in soil is critical to water resource management, nutrient management and contamination risk evaluation. Early research found that soil water flow takes place uniformly and, therefore, developed the concept of piston flow migration of soil water [4–6]. However, subsequent studies revealed preferential flow in soil, where a portion of water and solutes move along certain pathways (like large pores and cracks) and bypass a fraction of the porous matrix [7,8].

Although the two flow patterns have been proposed, it is difficult to differentiate preferential flow from piston flow via direct measurement of the changes in soil water content. Instead, tracers like Cl⁻ and water isotopes (²H, ³H and ¹⁸O) are used to assess the ratio of piston flow to preferential

flow [9–13]. But even so, it is still a great challenge to accurately separate the two forms of water flow under field environments, especially over long time scales, due to the small variation of soil water content. Moreover, most previous studies on water flow mechanisms are focused on shallow soil, and consequently, the water movement patterns in deep soil remain poorly understood. Recent progress on ecohydrology demonstrated that deep soil water extraction is a widespread phenomenon in forest ecosystems [14–16]. In this context, clarifying the water flow mechanisms in deep soil is important to ecohydrology [17].

Unlike shallow-rooted plants, deep-rooted trees are normally characterized by high water demand [18]. This stimulates trees to extract more water in deep soil [19,20], and finally results in the desiccation of deep soil, which occurs in both arid and humid regions [18,21–26]. Moreover, in water limited environments, the desiccated deep soil can hardly be replenished during the life span of the trees [16]. Consequently, trees are likely to experience more water stress, and are more vulnerable to disease, eventually leading to disease-induced death or being felled (deforestation) due to poor productivity. Moreover, in soil of a felled forest, the upper several meters of the soil profile normally contain higher $Cl^-$, due to, for example, the intensive application of chemical fertilizers, which is especially the case in the Loess Plateau [2].

Land use change from deep rooted trees to shallow rooted croplands generally results in reduced evapotranspiration and enhanced infiltration [27–30], which can be viewed as a recovery process of the desiccated deep soil. In this study we hypothesize that the intensive infiltration and the initial high $Cl^-$ concentration in desiccated soil have great advantages for the study of the water movement mechanisms due to: (1) just after deforestation, soil water content in deep soil is at the lowest stage, and therefore the soil water in this stage is in small pores due to the large binding force of the small pores [31]. In this context, nearly all of the $Cl^-$ should also be in these small pores. (2) During the recovery stage, the infiltrated water was in big pores and water flow rates were higher than in the small pores. Further, the small pores were filled with water containing high concentrations of $Cl^-$. (3) Collectively, in this study, during the water recovery process of deforested forests, the migration of $Cl^-$ was used to represent piston flow (it has the opportunity and time to mix with water containing high concentrations of $Cl^-$ in small pores), and the water that moved (in large pores) faster than the $Cl^-$ (in small pores) was defined as preferential flow (it does not have the opportunity or time to mix with water containing high concentrations of $Cl^-$ in small pores). Therefore, in this study, preferential water flow results from water velocity difference between pore and Darcy scales [8,31,32], which may have led to the separation of the $Cl^-$ migration front (moving in small pores) and the water moving in large pores (preferential flow path).

To test our hypothesis, four deforested apple orchards with a deforestation time ranging from 3 to 15 years were selected on the Loess Plateau. In each of the four selected orchards, there was a standing orchard that was planted at the same time as the deforested one, and therefore the standing orchard was used to benchmark the initial $Cl^-$ and soil water profiles of the deforested orchard. According to the maximum infiltration depth, soil water and $Cl^-$ were measured to a depth of 10 to 13 m to reveal the water flow pattern in deep soil. The results of this study are projected to promote our understanding of the water movement mechanisms in deep soil.

## 2. Materials and Methods

### 2.1. Site Description

This study was conducted in Changwu County (35°12.701′ N to 35°16.717′ N), Shaanxi Province, China (Figure 1). All sampling sites were located on flat tableland with elevations of around 1200 m above sea level. The climate is semi-humid with an annual precipitation of 580 mm, 70% of which falls from June to September [33]. Rainfed agriculture is the dominant production system. However, since the 1980s, apple trees were widely planted. By 2000, apple orchards covered 60% of the arable land in Changwu County [34]. As one of the first areas in China to grow apples extensively, old apple

orchards were found facing low soil water content in deep soil layers, and some of them have been cut down in recent years due to poor productivity. Therefore, this area is ideal for studying the movement of soil water following deforestation.

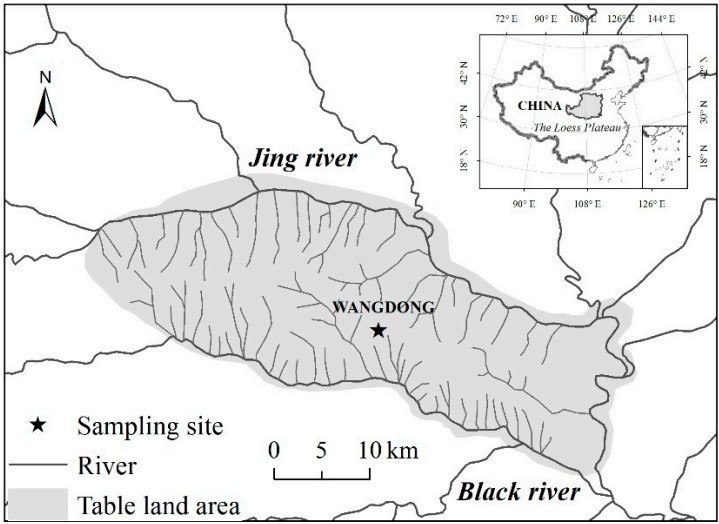

**Figure 1.** The locations of sampling sites.

### 2.2. Soil Sampling and Analysis

In May 2015 (before the rainy season), four croplands derived from deforested apple orchards were selected around the Changwu experimental station (Figure 1). The deforestation age of the selected sites ranged from 3 to 15 years. In each site, there was a standing orchard that was planted at the same time as the deforested one, and therefore the standard orchard was used to benchmark the initial $Cl^-$ and soil water profiles of the deforested orchard. Based on the maximum infiltration depth of the deforested sites, soil cores were taken using a soil auger to a depth of 10 to 13 m in each paired orchard and cropland (6 cm in diameter). Soil cores were collected at 20 cm intervals. The gravimetric soil water content (SWC) of each segment was measured using the oven drying method and subsequently converted to volumetric soil water content. To determine $Cl^-$ concentrations in soil water, 25 mL of deionized water was added to 5 g of the oven-dried soil sample, then agitated on a reciprocal shaker table for 1 hour, and centrifuged at 5000 rpm for 30 min. The concentrations of $Cl^-$ in the supernatant solution were analyzed by ion chromatography (Dionex ICS-1100, Thermo Fisher Scientific). $Cl^-$ concentrations in soil water were recalculated by:

$$Cl_{sw} = \frac{Cl_{ex} \times 5}{\theta} \tag{1}$$

where $Cl_{sw}$ is $Cl^-$ concentration in soil water (mmol $L^{-1}$); $Cl_{ex}$ is $Cl^-$ concentration in the extracted supernatant (mmol $L^{-1}$); 5 is the ratio of water to soil used in the extraction process; $\theta$ is the mass water content of soil (g $g^{-1}$).

### 2.3. Calculation of the Replenishment of Depleted Deep Soil Water

It takes a long time to directly observe the water migration process since deforestation. Instead, the space-for-time substitution method [35] was used in this study to detect the long-term water flow mechanisms in deep soil. Compared with the standing orchard, soil water changes in the deforested

sites were used to investigate the long-term water movement in deep soil. In this context, the total replenishment of depleted soil water storage (TR, cm) was calculated by:

$$TR = \int_0^L (\theta_{t2} - \theta_{t1})dz \tag{2}$$

where $\theta_{t2}$ is the soil water content in the deforested orchard (m$^3$ m$^{-3}$); $\theta_{t1}$ is the soil water content of the corresponding standing orchard (m$^3$ m$^{-3}$); and L is the depth of soil profile (cm).

### 2.4. Quantify Piston Flow and Preferential Flow Using Measured Cl⁻ and Soil Water Profiles

At the initial stage after deforestation, deep soil is desiccated and therefore nearly all the water and Cl$^-$ are concentrated in small pores. During the recovery process, the infiltration water ("newer" water) should be in pores larger than the "older" water filled pores (with high Cl$^-$ concentrations). As large pores are normally featured with a fast water flow velocity, herein, we use the downward migration of Cl$^-$ to represent piston flow, and the water moving faster than Cl$^-$ to derive preferential flow (Figure 2). In this process, the deforested site had the same Cl$^-$ input time as the corresponding standing orchard. Considering that multiple Cl$^-$ fronts will be formed due to the unstable input of Cl$^-$ in the orchard, we took the midpoint of the lowest dispersion front as the reference point to calculate the Cl$^-$ migration distance. The piston flow amount (cm) was calculated according to the migration distance of the reference point:

$$PF = 1 \cdot \bar{\theta} \tag{3}$$

where PF is piston flow (cm), l is the moving distance of the reference point (cm), and $\bar{\theta}$ is the average soil water content (cm$^3$ cm$^{-3}$) across the migration zone between the two reference points.

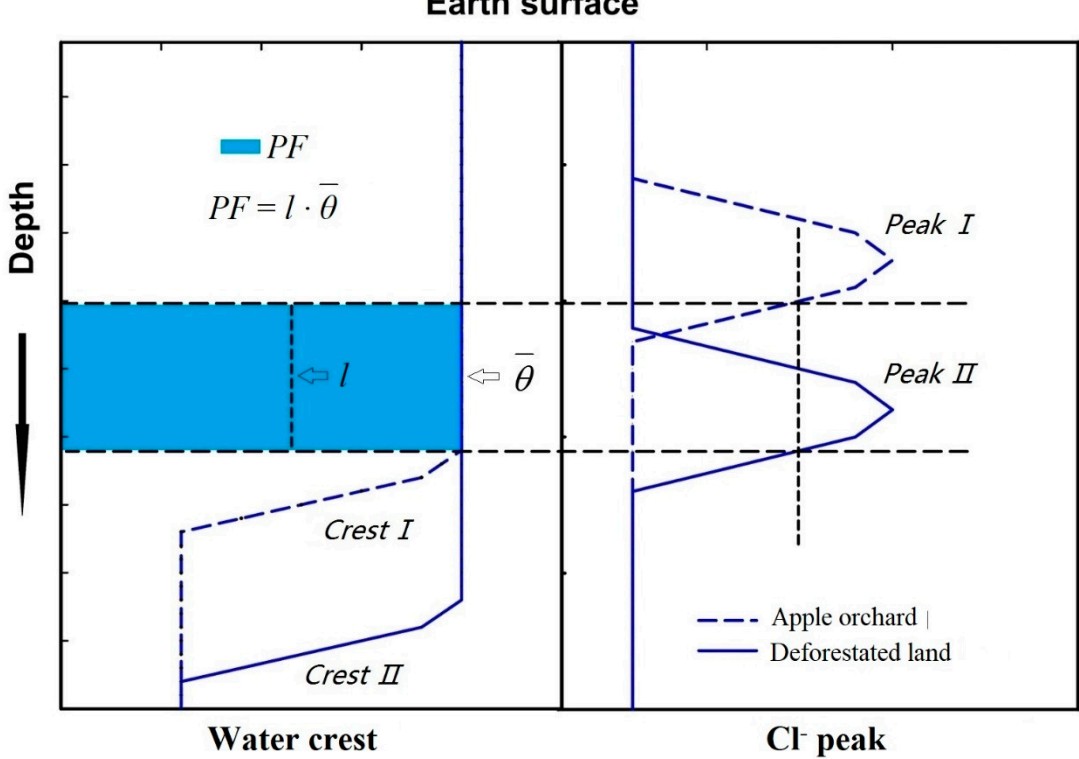

**Figure 2.** Diagram of water movement accompanying Cl$^-$.

Preferential flow amount (cm) was calculated based on soil water storage increase beneath the reference point:

$$P_rF = \int_{Z_1}^{Z_2} \theta dz \tag{4}$$

where $P_rF$ is preferential flow (cm), $Z_1$ is the depth of the reference point after recovery (cm), and $Z_2$ is the maximum infiltration depth (cm).

## 3. Results

### 3.1. The Evolution of Cl⁻ Profiles following Deforestation

In all the measured profiles, there were large $Cl^-$ concentration peaks (Figure 3) with $Cl^-$ concentration ranging from 9 to 48 mmol $L^{-1}$ (Figure 3). In some profiles, more than one peak appeared, which may be attributed to unstable fertilization after afforestation. The $Cl^-$ concentration stabilized below the crest area, with concentrations ranging from 0.25 to 1.43 mmol $L^{-1}$, which was far less than in the above strata. Moreover, in all paired sites, the shape of the $Cl^-$ profile was almost the same between the recovering site and the standing orchard. This result demonstrated that in each paired site, spatial variability was quiet small, and therefore, the appeared difference between the $Cl^-$ peak and soil water can be used to investigate water flow mechanisms in deep soil.

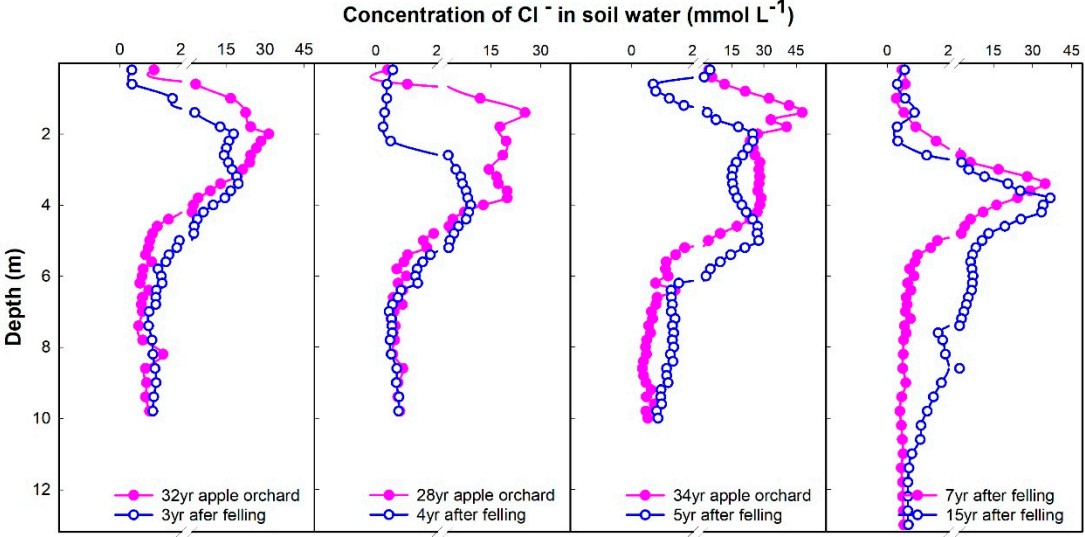

**Figure 3.** Concentration and storage of $Cl^-$ in soil water of 4 recovery sites.

### 3.2. The Amount of Piston Flow Tracked by Cl⁻ Migration

To accurately determine the migration distance of the $Cl^-$ crest, the concentration of $Cl^-$ in each soil layer was divided by the maximum value of that profile (Figure 4). Based on the relative concentration profiles, the midpoint of the lowest front moved 70, 80, 100 and 70 cm for site 1, 2, 3 and 4, respectively (Figure 4 and Table 1). Accordingly, during the water recovery period, the piston flow amount was 109.8, 123.6, 154.7 and 165 mm, for site 1, 2, 3 and 4, respectively (Table 1).

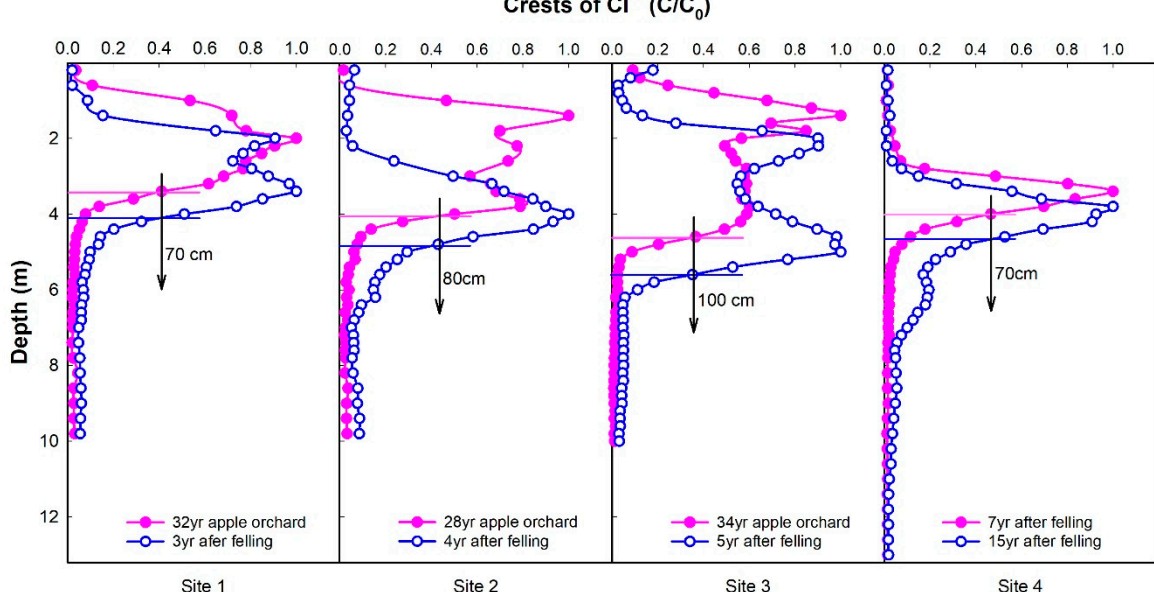

**Figure 4.** Movement of chloride crests in soil water.

**Table 1.** Soil water recharge calculated according to the movement of chloride.

| Variable | Recovery Time (year) | | | |
|---|---|---|---|---|
| | **3 (Site 1)** | **4 (Site 2)** | **5 (Site 3)** | **7–15 (Site 4)** |
| MD | 0.7 (3.4 to 4.1 m) | 0.8 (4 to 4.8 m) | 1 (4.6 to 5.6 m) | 0.7 (4 to 4.7 m) |
| PF | 109.75 | 123.64 | 154.68 | 164.97 |
| APF | 36.58 | 30.91 | 30.94 | 20.62 |
| $P_rF$ | 107.42 (4.1–7.3 m) | 117.24 (4.8–7.6 m) | 80.73 (5.6–8.4 m) | 307.86 (4.7–10 m) |
| ($P_rF$ + PF) | 217.17 | 240.88 | 235.41 | 472.83 |
| $P_rF$/($P_rF$ + PF) | 49.46 | 48.67 | 34.29 | 65.11 |

MD: Moving distance of chloride crest, m; PR: Piston recharge accompanying chloride crest, mm; APF: Average annual piston recharge, mm $yr^{-1}$; $P_rF$: Preferential flow below chloride crest, mm; ($P_rF$ + PF): Total recharge below the reference point of the $Cl^-$ front before recovery, mm; $P_rF$/($P_rF$ + PF): The percentage of preferential flow in total recharge, %.

### 3.3. The Amount of Preferential Flow Derived from Soil Water Increase below the $Cl^-$ Front

In the deforested sites, although $Cl^-$ migrated to deeper strata during water recovery processes, an evident increase in soil water content was observed below the $Cl^-$ front (Figure 5). The maximum water infiltration depth, on average, was 2.8~5.3 m ahead of the depth of the $Cl^-$ dispersion front, indicating the existence of preferential flow. According to Equation (4), the amount of preferential flow was 107, 117, 81 and 308 mm for site 1, 2, 3 and 4, accounting for 49.46%, 48.67%, 34.29% and 65.11% of the total recharge, respectively (Table 1).

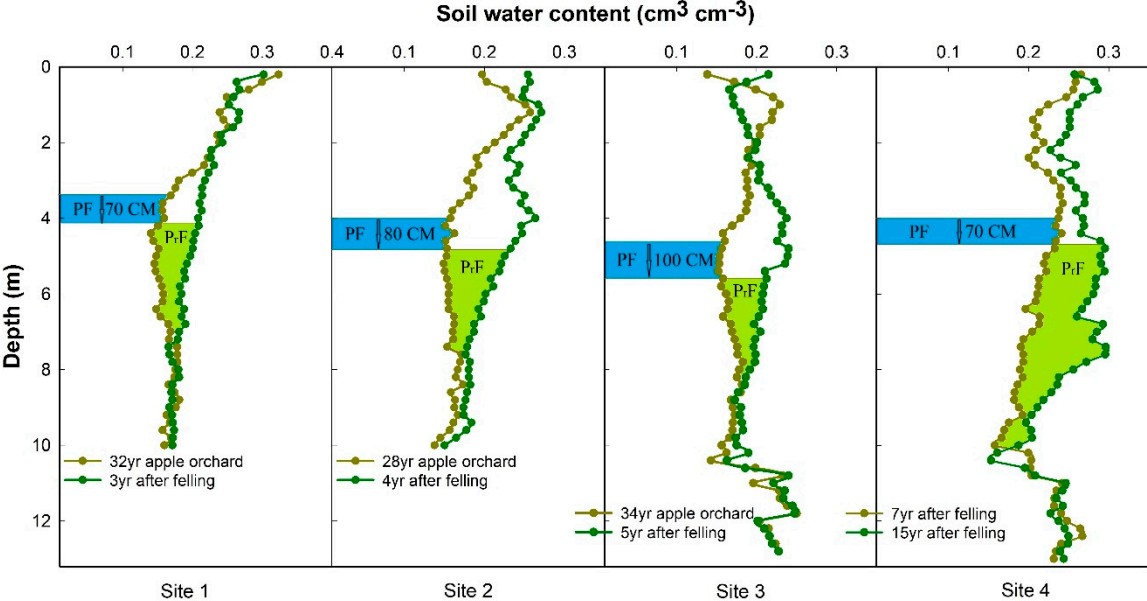

**Figure 5.** Soil water content, PF and $P_rF$.

## 4. Discussion

### 4.1. Why Do Piston and Preferential Water Flow Simultaneously Appear in the Deep Strata of Loess during the Water Recovery Process?

It is well known that water transport in soil is heterogeneous, while most previous research on water flow mechanisms are focused either on soil columns in the laboratory [36,37] or on shallow soil in the field [38–40]. In addition, most of these studies were conducted at/near saturated soil due to the long time need to detect unsaturated water flow [8]. To date, little data are available to quantify piston and preferential water flow in deep soil. In this study, using the space-for-time substitution method, we found that although $Cl^-$ migrated to deeper strata with infiltration, an evident increase in soil water content was found below the $Cl^-$ front, indicating the presence of both piston and preferential water flow in the deep loess (Figure 5).

The presented preferential water flow in this study is far different from previous studies and can be summarized by the following three aspects: (1) in this study, for all sampling sites, preferential flow appears in soil cores of only 4 cm in diameter. This result suggests that the preferential water flow in this study is, to a large extent, a widespread phenomenon. This is different from previous finger flow and macropore flow, where water only migrated in a small part of the soil matrix [41–43]. (2) The preferential water flow velocity in this study is far smaller than most previous reports. In this study, the average water velocity was only 125–243 cm $yr^{-1}$, while in previous reports, the pore water velocity of preferential water flow could be greater than 7000 cm $y^{-1}$ [44]. The smaller water velocity is mainly attributable to the low soil water content under field conditions, which ranged from 0.158 to 0.207 $cm^3$ $cm^{-3}$ below 3 m before recovery. (3) The presented preferential water flow in this study appeared in deep soil (from 7.3 to 10 m), while most previous studies reported preferential flow in shallow soil [13].

Evidently, pore water velocity difference remains key to the appearance of preferential water flow. Pore water velocity can be caused by differences in soil properties at both pore scales [8,31], and can also be caused by unstable "fingers" by large structural pores in coarse soils [45,46]. In this study, preferential water flow was observed in the deep loess following deforestation. The appearance of this type of preferential flow may be attributed to the following reasons: (1) in the deep soil of old apple orchards, soil water content has close to its lower limit of root water uptake due to intensive water extraction after afforestation [16]. In desiccated deep soil, water and dissolved $Cl^-$ are mainly concentrated in pores with small diameters and slow pore water velocity [31,41]. (2) After deforestation,

due to the small evapotranspiration rate of shallow rooted croplands, a relatively large amount of water infiltrated to deep soil, resulting in greater water pressure gradient around the infiltration front. Moreover, the infiltrated water was in pores with a larger diameter than the water/Cl$^-$ filled pores just after deforestation. Therefore, during the water recovery process of the deep loess, pores with high Cl$^-$ concentrations moved more slowly than the new infiltrated water, resulting in the appearance of preferential flow.

### 4.2. The Implications of Two Different Flow Patterns in Deep Soil on Ecohydrological Processes

Preferential flow significantly impacts ecohydrological responses to precipitation, and therefore, regulates both groundwater and surface water quantity and quality [8]. Preferential water flow with a high pore water velocity can hardly be extracted by plant roots before it is converted to streamflow and groundwater recharge, resulting in the ecohydrological separation of water between trees and streams [47]. Preferential water flow has been reported as an important channel for groundwater recharge, and can even contribute to more than 80% of total recharge [48]. Our study demonstrated that preferential water flow not only happens in shallow soil, but also appears in deep soil under natural conditions, where pore water velocity was small (Figure 5). This result suggests that chemical tracers, like Cl$^-$, cannot detect all infiltrated water. Therefore, previous estimations of groundwater recharge based on the Cl$^-$ mass balance method might have considerably underestimated groundwater recharge [49,50].

## 5. Conclusions

Based on measured Cl$^-$ and soil water profiles in four paired sites, this study quantified long-term piston and preferential water flow in the deep loess. For all sites, an evident increase of soil water was found below the downward Cl$^-$ front, demonstrating the existence of preferential water flow in the deep loess under field conditions, where pore water flow velocity was small. According to our measurements, preferential water flow can account for 34–65% under natural conditions. Our study revealed the widespread characteristic of preferential water flow and advanced our understanding of water flow mechanisms in deep soil.

**Author Contributions:** Conceptualization, Z.Z., B.S. and M.L.; writing—original draft preparation, Z.Z.; writing—review and editing, Z.Z., B.S., M.L. and H.L.

**Funding:** This research was funded by the National Natural Science Foundation of China, grant number 41630860, 41877017 and 41601222, the Fundamental Research Funds for the Central Universities, grant number 2452017317, and the 111 Project, grant number B12007.

**Acknowledgments:** We appreciate the technician Jingjing Jin from the Key Laboratory of Agricultural Soil and Water Engineering in Arid and Semiarid Areas, Ministry of Education, Northwest A&F University, for her assistance on the instrument used. The reviewers and editor provided valuable suggestions and comments to improve the paper greatly.

**Conflicts of Interest:** The authors declare no conflicts of interest.

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
