# Peer review of "Quantify Piston and Preferential Water Flow in Deep Soil Using Cl− and Soil Water Profiles in Deforested Apple Orchards on the Loess Plateau, China"

_water, doi:10.3390/w11102183_

Round 1
Reviewer 1 Report
The English should be improved.
The text contains several errors that should be carefully corrected, for instance, line 68 piton flow, figure 2 -slutes crest , line 151 concentration and so on.
Lines from 69 to 73 (which ae key lines) are not very clear, they nedd to be rephrased.
Reviewer 2 Report
The paper entitled” Quantify piston and preferential water flow in deep soil using Cl and soil water profiles in deforested apple orchards on the Loess Plateau, China” is generally well written. The only problem for me is the quality of the figs for this study. Please provide high-quality figs for publication purposes.
Reviewer 3 Report
Dear colleagues! I recommend your paper for publication because it contains unique and interesting data for hydrologists on the dynamics of water in deep soil. However, I am completely unsure of the correctness of your hypothesis about preferential flows. Loesses are homogeneous systems with strong capillarity, and capillarity neutralizes the finger effect, according to the Raats criterion, (Raats, 1973 (equation 8)). The differences you observed in the profile distributions of chloride tracers before and after deforestation can be explained without preferential flows. For example, the elimination of upward flow to the roots of trees, resisting the downward mass transfer of the chloride tracers. Or taking into account the effect of capillary rupture, established by Russian gyrologists using a similar chloride tracers in the 60-s of XX century (Rode, 52, see the attachment file 2, Chapter V, especially pp. 247-267, and as a similar field example after deforestation, see Yang Wen-Zhi experiment in p. 259). But this is a matter for your future research. Now I recommend a paper with a minor revision. All comments are given on the text. The most important of them is a clear description of the method of recalculation of the chloride tracer concentration in the soil water, and not in the soil, as you determined experimentally using a water extract 5:25 or 500% water content.
Raats P.A.C. Unstable wetting fronts in uniform and nonuniform soils // Soil Sci. Am. Proc. 1973. V.37. P.681-685. Rode. A.A. Soil moisture. M. Publishing House of the ANSSSR. 1952. 456 p. (In Russian, see the attachment file 2)
Sincerely Yours, Reviewer
P.S. Since I can not attach more than 1 file, I will ask the editors to send you the book AA Rode separately
